# Patient–Physician Interaction and Trust in Online Health Community: The Role of Perceived Usefulness of Health Information and Services

**DOI:** 10.3390/ijerph17010139

**Published:** 2019-12-24

**Authors:** Yuxin Peng, Pingping Yin, Zhaohua Deng, Ruoxi Wang

**Affiliations:** 1School of Medicine and Health Management, Huazhong University of Science and Technology, Wuhan 430030, China; 2Hubei Sanning Chemical Industry Company limited, Zhijiang 443200, China; yin_pp79@163.com

**Keywords:** online health information, patient–physician interaction, patient–physician trust, perceived usefulness

## Abstract

Background: In recent years, China has witnessed a surge in medical disputes, including many widely reported violent riots, attacks, and protests in hospitals. Asymmetric information between patient and physicians is one of the most critical enablers in this phenomenon, but the Web has become the primary resource for Chinese Internet applications to learn about health information and could potentially play a role in this pathway to patient–physician interaction and patient–physician trust. While considerable attention has been paid in some countries, there are few researches about China’s situation for this issue. The purpose of this quantitative study was to examine the influence of online health information and the online guidance of doctors in patient health information literacy on patient–physician interaction and patient–physician trust in China. Methods: A web-based survey was conducted to collect data from online applications with health problems. A structural equation modeling was used to analyze the data to test the hypotheses. A total of 446 participants from the Tongji Hospital in Wuhan and Huazhong University of Science and Technology hospital participated in the study. Results: Our analysis shows that the usefulness of online health information and the online guidance of doctors both significantly influence the trust of the patient toward physicians and interaction with physicians. Furthermore, the patient–physician interaction also has a significant impact on the patient–physician trust. Conclusions: There are many studies on the influence of online health information on the doctor–patient relationship, whereas a little research has examined this relationship between health information online support from doctors and patient–physician interaction by quantitative empirical analysis. This study also explores the online guidance role of doctors and whether doctor–patient communication will affect the trust of doctors and patients. The practical implications of this study include an improved understanding of the function of online health information and potential impacts regarding the interaction with physicians and trust toward physicians that can be used to resolve conflicts between doctors and patients.

## 1. Introduction

According to the statistics released by the China Internet Network Information Center (CNNIC), as of February 2019, the total number of Internet applications in China is 829 million, and the Internet penetration rate reached 59.6% [1]. Internet applications who have browsed online health communities (e.g., 39 Health Network, Sina Health, Sohu Health), and who have used search engines such as Baidu or Google to learn about health information accounted for 75.6% and 69.8%, respectively [2]. Compared with the traditional way of consulting with doctors or relatives or friends, Chinese Internet users are more inclined to use Internet applications to obtain health information. Studies have shown that patients’ access to health information through the Internet has a positive impact on the improvement of patient–physician relationship [3,4,5,6]. The Internet has become the central resource for the public to obtain health information in China.

Some studies found that 74.9% of medical staff viewed the current patient–physician relationship with a negative attitude, and medical disputes have frequently been reported such as violent riots, attacks, and protests in hospitals in China [7]. The tense patient–physician relationship mainly results from the lack of trust between doctors and patients, communication difficulties [8], asymmetric information [9], and cost problem [10]. It can be said that the contemporary doctor–patient relationship in China is still not optimistic.

Health information refers to the sum of health-related information or disease-related information. Patients who have a good grasp of health knowledge and their own health status have highly compliance [11] and actively participate in the whole process of diagnosis, prescription, and treatment [12]. From the perspective of social support, online health information support can guide patients to better self-management and self-care [13]. In addition, online health information not only can strengthen patients with increased knowledge, but also has the potential to foster greater patient engagement and decision-making in health maintenance and care [5,14,15]. The report shows that 45% of patients access health information online to understand illness symptoms description, basic characteristics of the disease before seeing doctors, so as to gain better communication in the process of medical treatment; 71.7% of them think it has a positive impact on their relationship with doctors [16]. This positive effect is stronger when patients and doctors discuss the information they find online [17]. Online health information makes patients more aware of "what to do," rather than just being told “how you should do” by doctors, improving patients’ sense of control and confidence in the medical process, according to a survey of cancer patients [15].

The medical information on the Internet is extremely rich, but it is difficult to identify effective information because of the insufficient medical knowledge of ordinary patients. Patients still have a large number of personalized health counseling needs between health information retrieval and offline treatment consultation, but online guidance from doctors such as online medical consultation services can supply the demand. Patients can obtain personalized health care consultation services with higher reliability provided by professional doctors [18]. Thus, we speculate that it is required for doctors to participate in the online health community or other social media to solve patients’ queries. However, little research focuses on the function of doctors’ online guidance for patients to acquire online health knowledge, and the influence on patient–physician interaction and trust. 

Patients discussing online health information with nurses and doctors during medical consultations have the potential to increase patients’ positive and knowledgeable engagement in health care decisions [19]. At the same time, online health information search has a positive impact on the co-creation of patient value in the process of medical treatment, which mainly promotes the interaction between doctors and patients in the process of clinical treatment and the decision-making of sharing, so as to improve the compliance of patients with medical guidance, which is the embodiment of patients’ trust in doctors [12]. The study found that the attitude of doctors in interaction has an important impact on doctor–patient trust, such as whether to answer patients’ questions about the side effects of drugs when prescribing, whether to focus on operation during surgery [20], and a good interactive process has a positive effect on enhancing doctor–patient trust. Therefore, we speculate that patients who perceived usefulness of online health information and online guidance from doctors will hold higher trust toward doctors when they are interacting offline.

Thus, we propose the following research questions:

Research question-How patients perceived usefulness of online health information and online doctors’ services affect doctor–patient trust?

To fill this gap, this paper undertakes an investigation on patients aiming at exploring the impact of the usefulness of online health information and the function of doctor’s online guidance on patient–physician trust and interaction. This study offers advice for physicians’ work and the findings also offer insights for hospitals regarding ways to engage the patients’ learning health knowledge on the web. And then, help to establish the harmonious relationship between patients and physicians.

## 2. Hypothesis Development

We divide the perceived usefulness of online health behavior into two aspect, usefulness of information and usefulness of service. Meanwhile, we propose that usefulness of information and service influence trust through interaction. The following are the proposed hypothesis.

### 2.1. Patient–Physician Interaction and Trust

Trust is defined as a belief that trustor will obey the act of the trustee when the information is insufficient, or it is difficult to make a reasonable decision for their self [21]. Doctor–patient trust is the foundation of clinical medicine, which can reduce the contradiction between doctors and patients, improve patient compliance, and promote the harmonious coexistence of doctors and patients. However, it is obvious that trust between doctors and patients in China is not enough [22], among which there are numerous factors that affect trust, such as the level of service in hospitals, the type of services provided, and the credibility of hospitals. Previous literature reveals that the outpatients have lower trust toward physicians than inpatients, because there is more interaction between inpatients and doctors, establishing a trust-based interaction between patients and physicians is important for patient compliance to treatment [23]. Owing to the lack of choice of therapeutic regimens’ opportunity, and short of health information sources channels, patients are at a disadvantage in the doctor–patient relationship. Now patients can make appointments with hospital experts on the Internet, and consult with experts who they trust and gain a better understanding. Patients can also learn about their condition through Internet consultation. They use the same treatment as other patients with the same disease to reduce panic and uncertainty, improving patient compliance, and thus have a positive impact on offline patient–physician trust [19].

In the interaction between physicians and patients, physicians not only convey professional knowledge by providing disease diagnosis to solve patients’ health problems, but also deepen the possibility of establishing a good personal relationship. According to social interaction theory, the interaction of doctors in solving patients’ health problem and providing emotional support to patients both significantly influence patients’ satisfaction and trust toward physicians [24]. The establishment of trust is based on significant interaction. Different types of patients may encounter various obstacles in nursing care, including external obstacles from family and internal obstacles from individuals, such as whether a patient with diabetes and gastroenteritis is willing to endure the abdominal cramps and receive nasal feeding tube to hold his nutrition [25]. Focus on the interaction of this kind of obstacles will improve the satisfaction of patients in treatment process, and enhance patients’ trust toward doctors. In our study, we argue that the patient–physician interaction after seeking online health information and perceived online help from doctors will have a positive significance to improve patient–physician trust. Therefore, we hypothesize as:

**Hypothesis** **1** **(H1).**
*Patient–physician offline interaction positively influences patient–physician trust in-person.*


### 2.2. Perceived Usefulness of Online Health Information

Perceived usefulness is defined as “the degree to which a person lies that using a particular system would enhance his performance.” Perceived usefulness is an important antecedent of behavior, such as consumer behavior [26], sharing behavior [27], adopting behavior [28], and using behavior [29]. In the commercial field, providing consumers with useful information can increase trust in products. 

The information adoption model first proposed the concept of “information usefulness,” which believed that the quality of information content and the credibility of the source directly affected the usefulness of seekers‘ perception of information [30]. There are many types of social media that can disseminate online health information, including search engines (e.g., Google, Baidu), portals (e.g., http://www.39.net/, https://www.iiyi.com/), Weibo, WeChat, forum (e.g., http://hbvhbv.info/), encyclopedia website (e.g., https://baike.baidu.com/, https://wiki.tw.wjbk.site/). You can link to other health websites through search engines. The portal contains a wealth of information, such as self-assessments of disease symptoms, medical guidelines, and drug introductions. Weibo can better spread health news. Forums are the main place for communication between patients. Encyclopedia website is a medical science popularization website with high authority.

There is a direct relationship between patients‘ perception of the usefulness of online health information and continuous use of health websites [31]. The search function in health websites replaces the roles of doctors and nurses [18] to a certain extent, thereby alleviating the trust crisis of offline patients. In addition, patient‘s subjective norms and trust in the platform will affect the use of online health platforms, and this effect is mediated by the usefulness of online information [32]. Studies show that the use of online health platforms can perceive doctors‘ performance during offline consultation (the quality of the doctor’s explanation and the length of communication), and a longer search on the Internet will reduce the impact of doctor performance on trust [33], so we predict that the usefulness of online health information can have a positive impact on doctor–patient trust.

Previous studies have found that patients can improve patient compliance by obtaining online health information [34], which is reflected in working with doctors, such as making corresponding diagnoses according to doctors’ instructions, following doctors’ orders, and improving health habits according to doctors’ requirements. It also reflects the good interaction between doctors and patients. In addition, the Internet-derived knowledge helps to communicate with doctors on an equal basis. Many patients usually get more health information through the Internet after seeing a doctor, which will help them interact with the doctor at the next visit [35]. Therefore, we can assume:

**Hypothesis** **2** **(H2).**
*Perceived usefulness of online health information positively influences trust in-person.*


**Hypothesis** **3** **(H3).**
*Perceived usefulness of online health information positively influences offline interaction.*


### 2.3. Perceived Usefulness of Online Doctors’ Service

In this study, patients get online doctors’ services through online doctor service platform (for example, https://www.haodf.com/,https://dxy.com/), including multiple forms, such as “graphic consultation service” in the form of pictures, words, and voice communication with doctors; one-to-one “telephone consultation” service in the form of one-to-one communication with designated experts by telephone, and some top doctors provide “fast phone” service (10% Call in the clock). The difference from general online health information is that patients and doctors communicate more authentically online. At present, relevant research on online consultation finds that patients pay attention to the value comparison with offline services when using online medical services. Only when online services meet users’ value expectations, users will have further offline medical behaviors [36]. In addition, patients’ attitudes and subjective norms toward online medical services will affect patients’ willingness to online behavior, and then affect the offline actual behavior [37]. The important antecedent of influencing attitudes is behavioral beliefs. In the field of electronic services, perceived usefulness is considered as an important behavioral belief [38]. Moreover, patients’ trust in doctors is directly related to users’ perception of distributive justice, procedural justice, interpersonal justice, and information justice of online medical services [39]. These justice are mainly reflected in the friendly interaction with doctors online, doctors’ interpretation of diseases, and presentation of diagnosis results. The above process is closely related to perceived service usefulness of doctors. Thus, we can hypothesize as:

**Hypothesis** **4** **(H4).**
*Perceived usefulness of online doctors’ service positively influences offline interaction.*


**Hypothesis** **5** **(H5).**
*Perceived usefulness of online doctors’ service positively influences trust in-person.*


The specific hypothesis model are shown in Figure 1.

## 3. Data Set and Variables

### 3.1. Participants and Data Collection

This study adopts a questionnaire survey to obtain data. We collected data for this study using survey method through Wenjuanxing, a platform providing functions equivalent to Amazon Mechanical Turk. The electronic questionnaire was disseminated and investigated on a variety of social media channels (e.g., WeChat, QQ, academic forum, online health community). The respondents covered multiple age groups. The questionnaires were distributed among Tongji Hospital in Wuhan and Huazhong University of Science and Technology hospital. As a result, we obtained a total of 500 responses. All of the responses were scrutinized, and those containing insincere or incomplete responses were discarded because too many values were missing. Finally, a total of 446 usable responses were obtained as data. Sample characteristics are shown in Table 1 below:

### 3.2. Measurement Instrument

To validate our research model, we used survey method in this study. The survey instrument was developed by adapting previously validated scales for our study. The questionnaire design is based on reference to existing data and theory in domestic and foreign literature. 

The items are measured by Richter’s five-grade scale, and the respondents are required to answer according to their own perception of online health information and doctor–patient interaction. All items were revised based on features of online health services, and were measured on five-point Likert scales, which range from “strongly disagree” to “strongly agree” and score from 1 to 5 correspondingly. 

Since the survey instrument is originally developed in English, we use back translation method to translate it into Chinese. The English instrument is first translated into Chinese by one of the bilingual authors whose native language is Chinese. Next, another bilingual author back translated the Chinese version into English. The two authors then compared the two English versions to check for inconsistency. A pretest was conducted on the survey instrument by interviewing eight experts in the area of information systems, medical informatics and health management. A total of 17 users of social media were also surveyed using the pretest. We revised the questionnaire based on the comments and suggestions received.

In addition to constructs in our studies, we also consider some control variables which have been used in literature including gender, age, and self-rated health status [40]. The specific measure of each variable is shown in Table 2.

## 4. Results

This study employed structural equation modeling using partial least square (PLS) analysis. As the second-generation multivariate causal analysis method, PLS can be applied to complex structural equation models and is less restrictive on sample size than other methods. Meanwhile, PLS is suitable for exploratory studies since it aims at theory building rather than theory testing [44]. The analysis was conducted by using SmartPLS 2.0. M3

We analyzed the reliability and validity using confirmatory factor analysis. As shown in Table 3, all Cronbach’s alpha and composite reliabilities are close to or above 0.7, thus demonstrating reliability for all constructs. The value of average variance extracted (AVE) of each construct is above 0.5 and item ladings are above 0.6, thus demonstrating good convergent validity. 

We also calculated the square root of each factor’s AVE and its correlation coefficients with other factors, and summarize the results in Table 4. Our results show that the square roots of the AVEs are all greater than the inter-construct correlations, thus demonstrating discriminant validity. Hence, we conclude that the quality of measurement model is adequate for testing hypothesized relationships.

PLS with bootstrapping procedure was used to test the hypothesized model. Estimates derived from the PLS analysis were used to test the research hypotheses. The results of the analysis are summarized in Figure 2. Patient–physician interaction significantly affect trust. Thus, H5 is supported. Usefulness of information is shown to significantly predict patient–physician interaction and patient–physician trust. Therefore, H2 and H3 are found to be true. The analysis results revealed that patients’ perceived usefulness of online service from doctor significantly influenced patient–physician interaction and patient–physician trust. Therefore, H4 and H5 are supported. All of the five hypotheses in our study are supported.

## 5. Discussions 

In this paper, we studied the effect of perceived usefulness of online health information and perceived usefulness of online service from doctor on patient–physician interaction and patient–physician trust. We also discussed the connection between patient–physician interaction and patient–physician trust. Through a survey method, we found that the perceived usefulness of online health information can urge patients to interact effectively with the doctors. Furthermore, we also discover that the effects of perceived usefulness of online service from doctor can boost the patient–physician interaction. Lastly, we confirm that the usefulness of patients’ perception of online health information and the usefulness of online services have a positive impact on patient–physician trust and patient–physician interaction during offline visits, and also confirm the positive impact of improving offline patient–physician interaction on offline patient–physician trust.

### 5.1. Implications

This study provides some interesting contribution to theory and practice. 

From the theoretical perspective, we explore the patient–physician interaction from two aspects, perceived usefulness of online information and usefulness of online service. Our empirical study confirms the effectiveness of that two components. Also, the two aspects and patient–physician interaction provide a deep insight into understanding the influence factor of patient–physician trust.

Second, this research enriches the theory of patient value co-creation. According to the theory of value co-creation, value is the realization of common interests by the participation of service participants in the process of resource integration. The behavior of patient value co-creation includes patient information search, information sharing, personal interaction, cooperating with diagnostic forces [45,46,47], complex with basis [48], complex with medical instructions [49,50,51]. This study plays an important role in realizing value co-creation in offline medical services by perceiving online medical experience. In addition, this study also enriches the related research of online and offline resource integration.

Third, compared with previous literature, we first consider the role of doctors in guiding the use of online health information for patients. Notably, our analysis uncovered that doctor’s help patients acquire the right, authoritative, and useful information on the Internet to facilitate interaction during the visit. Meanwhile, patients’ perceived usefulness of doctors’ online service would also enhance patients trust toward doctors.

From a practical perspective, this study suggests that improving patient–physician interaction and patient–physician trust should be combined with patient’s efficient use of online health information, such as improving the practicability and credibility of online health information, and doctor’s guidance and correction of information from other sources. Therefore, despite users get many sources of health information, but compared with the relatives and friends, Internet and social media, the advice and knowledge provided by medical professionals is still the most trusted and essential source of information that users believe [52]. China promulgated the regulations on the administration of Internet information retrieval services in June 2016, which holds that service providers have the primary responsibility for regulating network health information. To enable patients to obtain more health information and efficiently accept useful health information, we can establish more specialized website of health knowledge to bring convenience for other population to receive health information and to help patients integrate multi-channel information and reduce the use of misleading information by patients.

Second, in the patient–physician relationships, doctors’ attitude to online health information seeking is critical. Other research shows that if doctors are supportive of patients’ access to health information through the Internet, it will promote patients’ access to information, actively communicate with doctors and increase trust between doctors and patients [53]. Online health information can improve patients’ right to know and speak, make more harmonious for patient–physician relationship, and it will also improve patients’ perception and evaluation of doctor–patient relationship. Therefore, compared with traditional offline health consultation, health care providers should be encouraged to respond to the questions positively from web search. In addition to improving the offline service provided by doctors, online health professional counseling services can be developed to play the crucial role of medical professionals in the access and use of medical information for the general public.

At last, health care providers and health educators can even refer our measurement scales to check the level of patients’ perceived usefulness of doctors’ online service, the effects of their interaction and trust with consumers in the visit, and also can check the level of perceived usefulness of health information in social media. 

### 5.2. Limitation and Future Research

There are some limitations in this study.

First of all, we have indeed identified several determining factors of patient–physician interaction and patient–physician trust, however, through other relevant studies founds that more factors await exploration, such as self-efficacy, perceived risk, perceived benefit, etc.

Second, our study can also combine some theories to study trust between patients and physicians, for example, social exchange theory, social capital theory, etc.

Third, the study should discuss the impact of participants’ educational background. Participants from different educational backgrounds have different cognitive levels of health information, so the perceived usefulness may be affected, which should be further discussed in the follow-up study.

Therefore, future studies can consider other factors and other theoretical perspectives to improve the validity of our study model.

## 6. Conclusions

This paper contributes to the literature on patient–physician interaction and patient–physician trust by investigation patients’ perceived usefulness of online health information and online doctor’s service. Our research has proved that online experience of patients has an important impact on offline medical treatment, which provides a new way to improve the relationship between doctors and patients. This result not only implies the factors that affect trust and interaction between doctors and patients, but also reveals that doctors could consider enhancing interaction to promote trust between doctors and patients. Theoretical and practical implications for using online health information and doctors’ online service to encourage patient–physician trust are provided.

## Figures and Tables

**Figure 1 ijerph-17-00139-f001:**
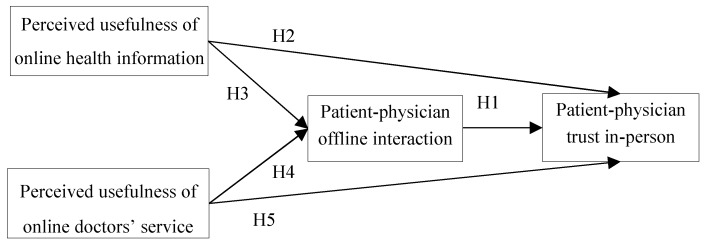
Research model.

**Figure 2 ijerph-17-00139-f002:**
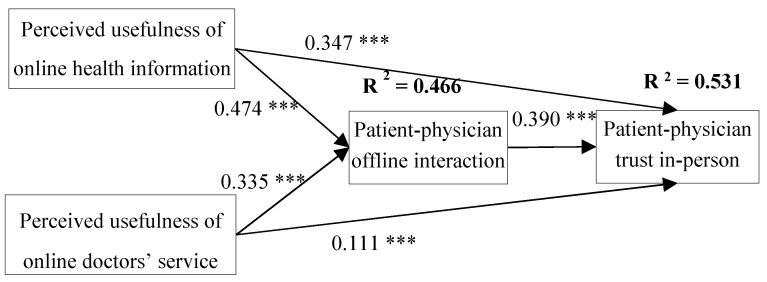
Analysis results of structural model. (*** Statistically significant at the 1% level; ** Statistically significant at the 5% level; * Statistically significant at the 10% level, two-sided test).

**Table 1 ijerph-17-00139-t001:** Demographic information.

Characteristics		Number	Percentage
Gender			
	male	222	49.5%
	female	224	50.2%
Age			
	≤20	18	4.0%
	20–30	288	64.6%
	≥30	140	31.3%
Self-reported health condition			
	poor	304	68.2%
	good	142	31.8%
intensify health consciousness			
	very helpful	110	24.7%
	helpful	308	69.1%
	no help	28	6.3%
online health information seeking			
	before visit	114	25.6%
	after visit	67	15.0%
	both	225	50.2%
	others	40	9.0%
source of health information			
	medical professional	173	38.8%
	online information	224	50.2%
	journal	11	2.5%
	friends	31	7.0%
	others	7	1.6%
Issues that should be improved			
	patient–physician communication	174	39.0%
	patient–physician trust	166	37.2%
	service attitude of medical professional	98	22.0%
	others	8	1.8%

**Table 2 ijerph-17-00139-t002:** Variable measurement and source.

Potential Variables	Items	Questions	Source
Perceived usefulness of online health information	1	Online health information helps me validate doctors’ diagnosis	Tan S S et al. [6]
	2	Online health information helps me realize my own health status	
Perceived usefulness of online doctors’ service	1	I receive caring and helping from doctors on the online health community.	Iverson S A et al. [5]; Oh, HJ et al. [14]
	2	Online doctors will offer me necessary information when I face disease-related problems.	
Patient–physician interaction	1	Interacting with doctors makes me feel smooth.	Oh, HJ et al. [14]; Ford S et al. [41]; Wu T et al. [42]
	2	I will try to get practical information from doctors for my disease management.	
	3	During my interaction with doctors, the doctors demonstrate sufficient devotion to the management of my problems.	
	4	When I find some information regarding my disease, I will bring them to my doctor.	
Patient–physician trust	1	I trust the doctors so much I always try to follow his/her advice.	Anderson L A et al. [43]
	2	My doctor is a real expert in taking care of medical problems like mine.	
	3	If my doctor tells me something is so, then it must be true.	
	4	I trust my doctor to put my medical needs above all other considerations when treating my medical problems.	

**Table 3 ijerph-17-00139-t003:** Item loadings and validities.

Construct	Items	Factor Loadings	Composite Reliability	Average Variance Extracted	Cronbach’s Alphas
Patient–Physician Interaction	INT1	0.715	0.846	0.579	0.853
	INT2	0.735			
	INT3	0.772			
	INT4	0.818			
Patient–Physician Trust	TRU1	0.772	0.8088	0.5153	0.844
	TRU2	0.758			
	TRU3	0.686			
	TRU4	0.648			
perceived usefulness of online health information	UI1	0.773	0.6985	0.5357	0.688
	UI2	0.691			
perceived usefulness of online doctors’ services	PH1	0.848	0.8326	0.7132	0.779
	PH2	0.841			

**Table 4 ijerph-17-00139-t004:** Correlation coefficient matrix and square roots of the average variance extracted (AVE) (shown as diagonal elements).

.	PUI	PUS	INT	TRU
PUI	0.732			
PUS	0.403	0.845		
INT	0.610	0.526	0.761	
TRU	0.633	0.455	0.662	0.718

Note: the square roots of AVE are in boldface. Abbreviations: PUI = perceived usefulness of information, PUS = perceived usefulness of service, INT = patient–physician interaction, TRU = patient–physician trust.

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
