# Peer review of "Patient–Physician Interaction and Trust in Online Health Community: The Role of Perceived Usefulness of Health Information and Services"

_ijerph, 2019, doi:10.3390/ijerph17010139_

Round 1
Reviewer 1 Report
Brief summary: This research sought to determine the relationship between patients' perceived usefulness of online health information and doctoral support, and the perceived quality of interaction and trust between them and doctors in in-person. Findings indicate that the positive association between these constructs exist, which might have an implication for theory and practice.
Broad comments:
Although this study tackled an interesting topic and used a reasonably large sample, several methodological and statistical concerns preclude the acceptance of the manuscript for the publication according to this reviewer's perspective. In addition, the presentation of the study's rationale and reporting of the obtained findings requires a major reconstruction and rewriting, taking into account the clarity of statements and grammatical punctuality.
The introduction requires large revisions in terms of text organisation and deletion of repetitive statements and ideas, as well as supporting the statements with references. Result reporting that includes causal inference should be omitted as it is a cross-sectional study, not longitudinal or with experimental design. It is possible that patients who trust their doctors and have a pleasant interaction will feel more confident in searching health information and support online, as Courtney et al. (2013) observed. Regarding methodological aspects, there are crucial information missing (see specific comments). With regards to statistical aspects - did the authors control for any demographic variable in their analysis, and health status as well? Did the authors ask participants how often did they visit the doctor (in any form)? Participants with impaired health will likely interact with a doctor both online and in-person, which is then a large confounding factor. Finally, in the Limitations section - how about mentioning that the cross-sectional study design precludes the causal explanation of the findings?
Specific comments:
Abstract - 446 samples or 446 participants in a sample?
P2 L42-43 - the latter part of the sentence provides a confusing statement - authors could paraphrase this
P2 L45-46 - this sentence seems unrelated to the rest of the paragraph as it refers to online health information - authors could place this in the previous paragraph or the one after the current
P2 L52 - how can "health information empower patients"?
P2 L53 - health information support online or in-person?
P2 L58-59 - how can it be affected, in which direction? The authors could clarify this thought and/or exemplify it.
P2 L61 - authors could replace "weak" with "insufficient"; also what does "ordinary patient" mean - and what would be an "unordinary patient"? This should be revised. The latter part of the sentence is redundant as authors mentioned already in the same sentence that it is difficult to recognise effective information due to lack of knowledge.
P2 L67 - "research" instead of "analysis"?
P2 L71-72 - what is provided to doctors from patients in this study, and in general, for the relationship to be perceived as "equal"?
P2 L72 - could authors provide an explanation of this theory so readers can be certain how does it align with the entire story?
P2 L76 - the mere presence of "online health information," or its perceived usefulness, or something else?
P3 L78-79 - this sentence provides little understanding of the concepts and relationships the authors aimed to examine - should be revised
P3 L80 - "effective interaction with doctors" online or?
P3 Fig.1 - authors should specify if they are referring to interaction and trust in the online environment or in-person
P3 L97 - why is it not enough, what is missing? Does the following sentence serve as an explanation of this observation? If so, it should be more clearly noted and/or sentences could be connected with linking words.
P3 L105-106 - this statement requires clarification and grammatical revision, and would benefit from referencing (especially regarding mentioned "panic" for which authors should be confident to mention here, but also consider the cyberchondria phenomenon that contrasts this statement)
P3 L107-109 - doctors solve patients' health problems in order to form a better relationship or vice-versa? The authors should reformulate this.
P4 L119 - positive "patient-physician interaction"?
P4 L121-123 - redundant paragraph as it is already discussed in previous paragraphs, hence it is unnecessary to define the concept at this stage - authors should either start the introduction with this or omit it completely.
P4 L129 - who are consumers in this context?
P4 L131-132 - should be grammatically revised as in the current form provides little sense.
P4 L43 and L44 - concepts in these hypotheses should be specified - patient-physician trust and interaction?
P5 L47-48 - authors should revise this sentence completely as it provides little sense in its current form.
P5 L149 - "mild patients" meaning patients with mild/less severe illness?
P5 L167 - Participants and data collection?
P5 L173 - "insincere responses" - how is this defined?
P5 L168-175 - several concerns: Did authors obtain ethics approval and did participants provided informed consent? How was the study aim presented to potential participants? Were they offered an incentive for their time devoted to this research? Did participants have the ability to terminate their participation at any moment? How long did the data collection last in average (eg 15 minutes)? Which social media channels were used - could authors name a few examples?
Table 1 - what were the mean age and the age range? Why only two categories for the self-reported health condition? "Journal" - an online journal or the printed version? How about other media such as TV?
P7 L185 - was the same response scale used in original instruments?
Table 2 - this is an uncommon way of presenting measurement instruments - questionnaires and their original developers should be named appropriately, alongside the description of what has been altered in the version utilised in this study.
P8 L202 - considering that the authors presented hypotheses for the research questions, as well as a proposed model emerging from the literature (Figure 1), why is the approach adopted here theory building and not testing? Could this be elaborated?
P8 L203 Cronbach's alpha is not a measure of reliability but internal consistency
P10 L239 - how is this effectiveness confirmed?
Reference: Courtney R Lyles, Urmimala Sarkar, James D Ralston, Nancy Adler, Dean Schillinger, Howard H Moffet, Elbert S Huang, Andrew J Karter, Patient–provider communication and trust in relation to use of an online patient portal among diabetes patients: The Diabetes and Aging Study, Journal of the American Medical Informatics Association, Volume 20, Issue 6, November 2013, Pages 1128–1131, https://doi.org/10.1136/amiajnl-2012-001567
Reviewer 2 Report
The subject of the study is relevant and current.
To evaluate the patient-medical interaction and trust in the online health community is important to generate new validated methods.
I suggest that authors should improve:
1. Add to the characterization of the sample, the amount of patient-doctor interaction of each volunteer. With this, to determine some application measure.
2. Review the format of several references, as suggested by this journal.
3. It presents 35 references, of which 10 are more than 10 years old. These references must be updated.
4. Highlight that in the Discussion only confront the results with 2 authors of 35 your references. This should be improved to increase the quality of the manuscript.
Reviewer 3 Report
Authors investigated the association of online health information and trust between patient and physician. Web-based surveys were conducted, and answers were analyzed using Partial Least Square (PLS) supported Structural Equation Modeling (SEM). Constructs' reliability and validity were evaluated using confirmatory factor analysis. The results supported authors' hypothesis.
However, the English writing made it very hard to understand. Intensive English editing is needed for this manuscript.
Furthermore, there is very little detailed information about the questions asked in the questionnaire, neither the social media channels used for the questionnaire. Both of these information will introduce bias to the study. Please provide more details in the revision.
The education background of the patients, and health problem of the patients were not provided. These two may also introduce bias to the study. Please discuss the bias and limitations of study design and result in the discussion section.
Please provide the references of the methodology used in this manuscript. For example, the references to PLS-SEM, and confirmatory factor analysis need to be given.
Reviewer 4 Report
This is a good paper containing interesting results which merit publication. For the benefit of the reader, however, a number of points need clarifying and certain statements require further justification. There are given below.
1.The English of your manuscript must be improved before resubmission. We strongly suggest that you obtain assistance from a colleague who is well-versed in English or whose native language is English.
2.In the Literature review and hypothesis development part, the hypothesis model is put forward firstly, which is not consistent with the procedure of deducing the relationship between the variables of the empirical model, so it is suggested to adjust.Also,there are few explanations of the rationale for the study design.
3.A hypothesis needs to be presented on the basis of literature study。In the process of derivation of each hypothesis, the literature support is very weak. It is suggested to supplement the literature and strengthen the literature type, not to talk to oneself.
4.What was the rationale for Key variables? For example:the relationship among the concepts of doctor-patient relationship, doctor-consumer interaction and Patient physician interaction has not been clearly explained. From the beginning to the end, these concepts are used in confusion.
5.Try to set the problem discussed in this paper in more clear,write one section to define the problem, It is not clear in this paper whether patients need to improve the relationship between two variables (online health information, online doctors' service) or doctors need to improve the relationship between two variables (online health information, online doctors' service), or whether both need to be concerned Two variables (online health information, online docs' service), please re comb their research objects.
6.The logical structure of the article is inconsistent in the overall design:The three questions put forward in line 75-80 page 2-3 in the Introduction are inconsistent with the five hypotheses in the Literature review and hypothesis development later,It is suggested to comb the consistent relationship between the three research questions and hypothesis questions.
Please add the related webpages of two variables (online health information, online docs' service) proposed in this paper, that is, please provide the website description in the relevant location to enhance the practicality and reality of the article. The statements on lines 232 to 234 on page 10 are contradictory. Please modify the logical relationship of this sentence based on the research results of this paper. As for variable 1, whether these two variables have direct or indirect influence, if there is indirect influence, this paper does not do intermediary analysis. If we want to reflect the indirect influence, we suggest to supplement intermediary analysis. Please show the version of the statistical software version used in this study. The software version number given in this paper is wrong(lines 202 page 10). The software version should be SmartPLS 2.0.M3. The content in Table 1. Demographic information is not only demographic data, but also variable content. It is recommended to modify. Please attach the questionnaire of this study. It is suggested to carefully check the accuracy of the references provided, for example: the month of line 38 on page 1 should be June, not August; the document data of line 43 on page 2 is 2010, but the actual text data is 2007; the data source of line 41 has no literature support, please supplement. The format of the text needs to be standardized, for example: there are numbers on the keywords, please check and modify. The data sources are inconsistent. The "Abstract" says that the data source is " A total of 446 samples from the Tongji Hospital in Wuhan and the school hospital of Tongji Medical College of HUST" (line 22-23, page 1), but (page 5, line 171). The part of data collection says that "the questions are distributed along multiple cities across the country." the data sources are inconsistent. This is A fatal problem. About Section 2, the author(s) not introduced the prior studies about variables. However, we could not find the definition of variables in this study. Therefore, please state clearly about variables in this article. Some critical statistical information such as the bootstrapping samples, Goodness-of-fitness (GOF) and Coefficient of determination (R-square) are suggested to show in the analysis result if necessary. Please try to illustrate the research limitations of this study. There is only one sentence to explain the limitations in this paper: "The results of this study should be considered, given its limitations. ".it is suggested that the author fully excavate the limitations of this paper.

Round 2
Reviewer 1 Report
Thank you for adequately addressing previously raised concerns.
Reviewer 4 Report
The author replied to all the 17 modification opinions put forward by the review and carefully revised them. The review found that they met the requirements and could be published!